# Revolutionizing Allogeneic Graft Tolerance Through Chimeric Antigen Receptor-T Regulatory Cells

**DOI:** 10.3390/biomedicines13071757

**Published:** 2025-07-18

**Authors:** Alvin Man Lung Chan, Rajalingham Sakthiswary, Yogeswaran Lokanathan

**Affiliations:** 1My CytoHealth Sdn. Bhd, 5th Floor, Plaza Hamodal, Lot No. 15, Jalan 13/2, Section 13, Petaling Jaya 46200, Selangor, Malaysia; alvinchan@cytoholdings.com; 2Department of Medicine, Faculty of Medicine, Universiti Kebangsaan Malaysia, Kuala Lumpur 56000, Malaysia; sakthiswary@hctm.ukm.edu.my; 3Department of Tissue Engineering and Regenerative Medicine, Faculty of Medicine, Universiti Kebangsaan Malaysia, Kuala Lumpur 56000, Malaysia; 4Advance Bioactive Materials-Cells UKM Research Group, Universiti Kebangsaan Malaysia, Bangi 43600, Selangor, Malaysia

**Keywords:** chimeric antigen receptor, T regulatory cells, immunosuppressive therapy, solid organ transplant, allograft

## Abstract

**Background/Objectives**: Organ transplantation is a life-saving intervention for patients with terminal organ failure, but long-term success is hindered by graft rejection and dependence on lifelong immunosuppressants. These drugs pose risks such as opportunistic infections and malignancies. Chimeric antigen receptor (CAR) technology, originally developed for cancer immunotherapy, has been adapted to regulatory T cells (Tregs) to enhance their antigen-specific immunosuppressive function. This systematic review evaluates the preclinical development of CAR-Tregs in promoting graft tolerance and suppressing graft-versus-host disease (GvHD). **Methods:** A systematic review following PROSPERO guidelines (CRD420251073207) was conducted across PubMed, Scopus, and Web of Science for studies published from 2015 to 2024. After screening 105 articles, 17 studies involving CAR-Tregs in preclinical or in vivo transplant or GvHD models were included. **Results:** CAR-Tregs exhibited superior graft-protective properties compared to unmodified or polyclonal Tregs. HLA-A2-specific CAR-Tregs consistently improved graft survival, reduced inflammatory cytokines, and suppressed immune cell infiltration across skin, heart, and pancreatic islet transplant models. The inclusion of CD28 as a co-stimulatory domain enhanced Treg function and FOXP3 expression. However, challenges such as Treg exhaustion, tonic signaling, and reduced in vivo persistence were noted. Some studies reported synergistic effects when CAR-Tregs were combined with immunosuppressants like rapamycin or tacrolimus. **Conclusions:** CAR-Tregs offer a promising strategy for inducing targeted immunosuppression in allogeneic transplantation. While preclinical findings are encouraging, further work is needed to optimize CAR design, ensure in vivo stability, and establish clinical-scale manufacturing before translation to human trials.

## 1. Introduction

Lifelong immunosuppression is necessary to prevent graft-versus-host disease (GvHD) following allogenic graft or organ transplantations. The lymphodepletion strategy has become the standard protocol for solid organ transplants and has been shown to improve overall quality of life [1]. However, the long-term survival of the transplanted graft cannot be guaranteed. Prolonged intake of conventional immunosuppressants is associated with several complications, with opportunistic infections being high on the list. Take, for instance, the frequently reported EBV-associated malignancy of post-transplant lymphoproliferative disorders (PTLD) [2]. To achieve a successful graft assimilation, there should be an ideal balance between the persistence of allogeneic transplants and the prevention of graft rejection, as illustrated in Figure 1. This biological revision should take place during the refractory period of a depleted patient’s immunity and contact with donor tissue antigens to allow the emergence and expansion of graft-tolerant immune cells. However, the absence of the required signalling factors and poorly regulated conditions has led to chronic or delayed rejection. Premature graft failure from hyperacute or acute rejection of the graft is prevalent despite highly stringent screening protocols in the selection of compatible donors.

Regulatory T cells (Tregs) are a specialized subpopulation of CD4^+^ T lymphocytes that are crucial in maintaining immune homeostasis and self-tolerance. These cells express high expression of CD25 (the α-chain of the IL-2 receptor) and low expression of CD127 (IL-7 receptor α-chain). More importantly, the transcription factor FOXP3 that enables Tregs acts as the immune system’s natural suppressors [3,4]. They exert immunosuppressive functions via multiple mechanisms, including the secretion of anti-inflammatory cytokines (IL-10 and TGF-β), dendritic cell modulation, deactivation of effector T cells (Teff), and cytolysis of activated immune cells. In contrast, Teff cells (Th1, Th2, and Th17) are responsible for executing immune responses by generating pro-inflammatory cytokines (IFN-γ, TNF-α, IL-2, and IL-17), aimed at clearing infections or target mismatched antigens [5]. However, in the context of transplantation or autoimmunity, unregulated activity of Teff cells often results in graft rejection.

Besides, conventional T cells (Tconv), which refer to CD4^+^FOXP3^−^ T cells, are the pool of naive and memory T cells that differentiate into Teff subsets during antigen stimulation [5,6]. These cells orchestrate immune responses against foreign antigens and are central to both protective immunity and immunopathology. Compared to Tregs, Tconv cells do not express intrinsic regulatory functions and can become activated and drive graft rejection via secretion of inflammatory mediators and recruitment of Teffs. Thus, a higher Treg to Tconv ratio is considered favourable for transplant outcome and immune tolerance [7].

Hence, Tregs possess the advantage as a platform to initiate suppression of autoreactive and alloreactive T cells. Tregs are also valuable because they offer a biologically compatible means of generating immune tolerance without entirely suppressing the graft recipient’s immunity, unlike conventional treatment with immunosuppressive drugs [8]. This selective ability makes them attractive for cell-based immunotherapies for transplantation. In therapeutic models, Tregs have been shown to prolong graft survival, reduce chronic inflammation, and promote immune tolerance, highlighting their potential as an alternative to existing anti-rejection solutions [9]. However, Tregs alone often do not generate a sufficient or timely response, leading to masked symptoms and eventual graft rejection.

Over the years, there has been tremendous development of chimeric antigen receptor (CAR) technology. The foundation of CAR technology lies in engineering patient or donor immune cells, typically T cells, to recognize target cells. This technology essentially utilizes the target specificity of antibodies along with the powerful effector functions of T cells [10]. Leveraging their natural tumoricidal efficacy, CAR-modified T cells have become a promising tool to combat various cancers. Currently, CAR T cells are under investigation for haematological cancers, leukaemia, and lymphomas by targeting the CD19 antigen exclusive to B cells. Besides, research is well underway for CAR T cells to target viral antigens in infected tissues [11].

In parallel to the anti-tumour role of CAR T cells, CAR-regulatory T cells (CAR-Tregs) were recently proposed as candidates that could modulate erratic immune cells against self-tissues or transplanted organs [12]. Naturally, regulatory T cells (Tregs) contribute to donor-specific transplantation tolerance while having much fewer adverse effects than non-specific or polyclonal Treg immunosuppression [13]. Previously, the co-administration of donor Tregs with the corresponding graft has been shown to prevent transplant rejection [14]. However, the number of antigen-specific Tregs is often low or eventually diminished post-infusion. Thus, the ideal model of CAR Treg should be able to elevate their antigen-specificity function while remedying the viability issues to improve the overall retention and performance of conventional Tregs.

CAR is a modular platform, first consisting of an extracellular region, the binding domain of an antibody towards a specific antigen (e.g., HER2 or CD19); a transmembrane, co-signalling domain of a hinge and/or stalk for receptor dynamicity and flexibility (e.g., CD8 or CD28); and intracellular signalling domain for metabolic activation (e.g., CD28 and/or CD3ζ). Beyond their basic construct, the CAR can be additionally modified through ‘armour’ molecules that protect or react opposingly to inhibitory factors (e.g., enriched anti-inflammatory cytokines in tumour microenvironment); over- or under-expression of activating and inhibitory receptors; enhanced lymphocyte homing capabilities (e.g., VCAM-1) and other deficiencies that can be compensated for [15]. Granted that there are many different facets possible through CAR technology, but T cells already possess a high metabolic niche, which has led to biological “exhaustion” or senescence [16]. Ultimately, there remains much work to be completed and optimised in CAR design.

In the existing literature, there is critically limited evidence on the clinical applications of CAR Tregs. There is ongoing research on the safety of prototypes and preclinical studies using animal transplant models. Therefore, this article aims to evaluate the current available data and discuss the role of modified CAR T cells as a treatment for alloreactivity and tissue transplant tolerance.

## 2. Materials and Methods

A systematic review was performed with adherence to the guidelines of the International Prospective Register of Systematic Review (PROSPERO; CRD420251073207). The Search keywords were selected from medical subject headings (MESH) available from PubMed, namely, (i) receptors, chimeric antigen; (ii) organ transplantation; and (iii) graft rejection. Unconventional terms or synonyms not registered as MESH were included in the search. Database access and bibliography retrieval from Scopus, PubMed, and Web of Science (WOS) were accessible by the National University of Malaysia (UKM). Only “research articles” or “journal articles” published in the last 10 years (2015–2024) were downloaded as bibliographies containing title, keywords, and abstract. The bibliographies were labelled appropriately according to the databases, date of access, and results. The duplicates were merged using reference software: Mendeley v1.19.8 (Elsevier, Amsterdam, The Netherlands). The primary screening was performed relevant to the titles, abstract, and keywords. Thereafter, full-text screening was performed according to set inclusion and exclusion criteria as described below. Inclusion criteria: (i) CAR T, (ii) GvHD or any tissue transplant, (iii) preclinical or in vivo model, and (iv) controlled experimental studies. Exclusion criteria: (i) non-CAR T, (ii) not GvHD or any tissue transplant, (iii) non-preclinical or animal model studies, and (iv) uncontrolled experimental study.

A total of 105 records were acquired from merging the three bibliographic sources: PUBMED (46), SCOPUS (28), and WOS (31). Thirty-two records were removed from merging record duplicates, yielding 73 individual records (Figure 2). The primary screening eliminated 23 records, leaving 33 eligible studies for full-text screening. Finally, seventeen accepted records were selected for data extraction and analysis. The text screening and consensus were performed by authors (A.M.L.C. and Y.L).

## 3. Results

### 3.1. Conventional Tregs (Tconv) Have Limited Suppression Capacity

Regulatory T cells or Tregs play a critical role in maintaining immune homeostasis and adjusting irregular immune activities. Tregs are responsible for mediating between multiple target cells, utilizing various methods including induced apoptosis, regulating cellular metabolic pathways, and promoting or inhibiting anti- and pro-inflammatory cytokines, respectively. Some of these mutually interacting cells include antigen-presenting cells (APC), dendritic cells (DC), effector T cells, macrophages, and more.

In most studies reviewed (Table 1), the groups employing naive (n-), polyclonal (poly-), non-specific (NT-), or untransduced (UT-) Tregs did cause harm in either allo- or xenograft transplant models, compared to the effector (Teff) or conventional T (Tconv) cells where deliberate cytotoxicity was simulated [14,17,18,19,20,21]. However, the overall immunosuppression capacity of these Tregs did not last very long due to diminishing signals, which eventually caused the graft to be rejected by the recipient. Although polyclonal Tregs could theoretically be considered for universal model therapy, their well-established off-target effects remain a legitimate concern for clinical and translational application. For example, Noyan et al., 2017 reported significantly increased delayed-type hypersensitivity (DTH) and the ear-swelling response by 70% (*p* < 0.01) in the UT-Tregs vs. CAR-treated group [17].

Unbiasedly, the overall performance of Tregs was greatly enhanced post-transduction regardless of the CAR model induced. The study by Boardman et al., 2017 reported that CAR Tregs prevented immune cytokine trafficking to the graft by greatly suppressing IFN-γ and promoting IL-10 secretion in situ (*p* < 0.05) [18]. Similar outcomes were reported in Boroughs et al., 2019, whereby CAR Tregs prevented Teff-mediated tissue infiltration and apoptosis, prolonging the graft’s survival in the skin xenograft model (*p* < 0.0001) [24]. Additionally, the bioluminescence imaging by Dawson et al., 2019 observed fewer keratinocytes and involucrin destruction, prompting conserved function of the graft in the A2-CAR Treg-treated mice (*p* < 0.001) [25]. Beyond the commonly employed skin graft models, similar results were recreated from more biologically relevant transplants, including pancreatic islet [23,29,32], heart [28,30], and even CD19- or BCMA-CAR T cells for targeting B-cell malignancies [32].

### 3.2. HLA-A2 as the Basis of Immune Tolerance

In the context of organ and tissue transplantation, HLA mismatch is the primary source between the donor and the recipient that could trigger graft rejection. The HLA-A locus, especially HLA-A2, is one of the most important determinants of transplant compatibility. The HLA-A2 is a globally common Major Histocompatibility Class 1 (MHC-1) molecule found in humans [22,35]. Among the hundreds of HLA-A2 variants, the HLA-A*02:01 is the most prevalent model to study immune cytotoxic responses to foreign particle elimination [36]. Since nearly 40 to 50% of the population expresses HLA-A2, this frequent recurrence makes it an ideal target for immunological studies and the development of therapies catered to a wider range of populations.

Targeting HLA-A2 specifically with CAR Tregs or other CAR-based models has allowed researchers to create immunotherapy models of tolerance where the recipient’s immune system could be retrained to accept the transplanted organ, even if it is bearing HLA-A2 that would otherwise be rejected. This specificity also helps develop the recipient’s immune system to selectively tolerate the donor’s graft without relying on generalized immune suppression strategies, as seen in Figure 3. Thereby, reducing the risk of infections and other systemic complications associated with broad immunosuppression medications.

Out of the 16 studies reviewed, 11 studies that selected HLA-A2 CAR Tregs successfully demonstrated excellent immune suppression without compromising the survival of the animals [14,17,18,22,25,26,29,30,31,33,34]. Compared to non-specific, control CAR Tregs such as HER2- or CD19-CAR Tregs, the HLA-A2 CAR Tregs explicitly demonstrated their antigen specificity in the presence of HLA-A2^+^ graft.

In other notable models to consider, Lee et al., 2022 explored the application of anti-C4d-CAR Tregs to counteract ABO-incompatible (ABOi) heart transplantation [28]. From their study, the treatment significantly prolonged graft survival (*p* < 0.05) compared to control CAR Tregs, validated by low inflammatory activity in sample histology and cytokine analysis (IFN-γ and TNF-α). Compared to T cell-mediated rejection patterns, antibody-mediated rejection (ABMR) remains culpable for blood-incompatible (ABOi) hyperacute or rapid transplant rejection. Therefore, strategies to overcome the challenge of ABO blood type barriers could hasten the waiting time for donor screening and broaden the therapeutic window for life-saving transplants. Till then, the current regime of strong immunosuppression enforces excess immune suppression and could likely cause secondary symptoms such as sepsis.

Even perfect ABO matching remains susceptible to the innate immunity mechanism, as HLA mismatch could still trigger NK cell-mediated rejection from the absent signal of donor MHC Class I molecules. Since autologous T cells have compromised immune function while allogeneic T cells risk interception by NK cells or cause severe GvHD, Degagné et al., 2024 explored immune cloaking methods by engineering B2M-HLA-E onto CB-011 (anti-BCMA) CAR T cells to enable safer and effective treatment for multiple myeloma [38]. The study reported significant evasion and survival from clearance by natural killer (NK) cells at nearly 60% (*p* < 0.05) more than the control group.

### 3.3. Hypo-Responsive and Metabolic Exhaustion in Tregs

Thematically, studies have frequently cited a pronounced decrease in the viability and function of Tregs post-infusion [22,31]. With CAR Tregs, a moderate degree of retention could be observed, but most of the transfused cells were lost after weeks. One of the predominant reasons for poor retention of the CAR Tregs stems from the interleukin-2 (IL-2) paradox. Principally, one of the Tregs’ immunosuppressive behaviours is to deregulate the pro-inflammatory cytokines, including interleukins (IL), interferons (IFN), and the tumor necrosis factor (TNF) family of proteins, resulting in lesser invasion of immune cells in situ. However, this becomes counterproductive as Tregs are devoid of specific cytokines, especially IL-2, which is among the essential stimuli for immune cell proliferation [39].

Of note, Rosado-Sánchez et al., 2023 reported that the engraftment of CAR Tregs could translate into the uptake of HLA-A2 into CAR Tregs via trogocytosis [31]. This exchange could encourage Ab-specific suppression and elimination of HLA-A2 availability, affecting overall Treg functionality and graft survival. To ensure the transplanted cells’ survival, the supplement of exogenous IL-2 or repeated dosing of CAR Tregs was suggested. However, pro-stimulus factors possess highly complex interactions that could affect the outcome of therapy and incite Tregs or T cell anergy.

The chronic activation of CAR Tregs in the presence of highly expressed auto- or alloantigens is an underrepresented aspect in numerous studies. Tonic signalling was described as a constant, diminutive signal in T cells under basic conditions [40]. Although no reactions typically come of it, the constant engagement of TCR emits low-level signals that cumulatively build fatigue in cells from various feedback mechanisms. Other than the immune surveillance in vivo, purposely designed cultured scenarios for ex vivo, clinical expansion of CAR T cells may also contribute to tonic signalling [41]. In a study by Lamarche et al., 2023, tonic-signalling CAR (TS-CAR) resulted in exhausted Tregs with significant metabolic, transcriptomic, and epigenetic alterations [21]. Although the in vitro assessment was determinably similar, the TS-CAR Tregs failed to operate in the GvHD animal model, resulting in morbidity and mortality rates similar to the untreated group and significantly (*p* < 0.01) worse than control or UT-Tregs.

Thus, ex vivo expansion strategies become critical, not only to ensure the process does not alter Treg phenotype, function, and cause exhaustion in the process of generating sufficient cell numbers for therapeutic purposes. Early methods rely on tissue culture flasks or well plates that are either bead-based (e.g., anti-CD3/CD28 beads) and/or cytokine-tailored (e.g., IL-2 with IL-15) into the growth media. Studies have shown that both methods involving prolonged, specific stimuli led to cellular exhaustion. Therefore, newer strategies such as the G-REX^®^ (Wilson Wolf Manufacturing, St. Paul, MN, USA) platform offer a more relevant culture system and microenvironment for cells to grow optimally [42]. A key characteristic of the G-REX^®^ system is the unique, gas-permeable membrane at its base, enabling superior nutrient flow and gas exchange. An exploration conducted in 2013 by Chakraborty et al. surmised that 1 × 10^9^ cells could be generated from the system in 21 days using one culture vessel [43]. The expanded Tregs expressed high CD25 and CD4 (>90%) and relatively stable FOXP3 (~69%), which were assessed in an in vitro and in vivo model of GvHD. With further optimization, such as the Gotti et al., 2021 manufacturing protocol, >30 × 10^6^ cells/cm^2^ could be generated in <11 days using a single unit of G-REX^®^ compared to the 24 days of seventy-two tissue culture vessels [44]. Although the cell surface marker did not report any significant differences, the tumoricidal effect of the T cells cultured in G-REX^®^ was superior to the flasks. This technology has marked the viability of GMP manufacturing and clinical applications for T cell-based therapies. Simultaneously, there are also other commercial alternatives, such as the protocol validated by Marín Morales et al., 2019 used to compare the efficacy of the CliniMacs Prodigy^®^ (Miltenyi Biotec, NRW, Germany) to the G-REX^®^ platform for Treg expansion [45]. The study was able to generate a similar yield (2 × 10^9^ cells) with >90% subpopulation immunosuppressive phenotype (CD4^+^CD25^high^FOXP3^+^), with the added benefit of an automated, closed system that is more befitting of GMP requirements. Albeit, the economic factor of the culture systems must be considered to ensure low-production cost and patient accessible funds for life-saving therapy.

### 3.4. FOXP3^+^: A Major Consensus of Treg Function

The forkhead box P3 (FOXP3) is considered a “master regulator” of Tregs as it controls nearly all development, function, and maintenance of the Tregs. Henceforth, the FOXP3 was regarded as a lineage marker, distinctly programming precursor T cells to differentiate into Tregs and not Teffs. In the absence of dysregulated FOXP3 expression, Tregs could either fail to develop or lose their functional ability to suppress exacerbated immune responses. The reviewed studies frequently correlate the success of transducing CAR into Tregs without compromising their innate immunosuppressive qualities mostly through FOXP3^+^ expression. Although other factors have a role in Treg’s metabolism, the relevance of FOXP3 was unmatched by even CD25, CTLA-4, or CD69 markers.

Based on the evidence presented, the FOXP3^+^ expression should be maintained or increased to higher expression levels after transduction, independent of transduction efficacy or viability presented in Table 2. Lamarche et al., 2023 demonstrated that loss of FOXP3^+^, although created from metabolic exhaustion, still led to poor engraftment and function of the CAR Tregs [21]. In fact, Tregs inhibitory receptors—the likes of PD-1, TIM-3, LAG-3, GITR, and 4-1BB—were significantly raised (*p* < 0.05). Vice versa, Henschel et al., 2023 modified a CAR vector to contribute more FOXP3 expression into Tregs, finding that the CAR Tregs had continued stability and growth even under severe inflammatory, limited IL-2, and acidified conditions [34].

The observed reduction in FOXP3 expression may be a cumulative effect of various stages in the CAR-Treg manufacturing process, including ex vivo expansion, post-transduction modifications, and functional assessment. Currently, there is no established consensus regarding the optimal time point for cellular characterization, whether it should be performed during upstream manufacturing or immediately prior to clinical administration. From a quality control and standardization perspective, earlier characterization is preferable. However, the temporal constraints inherent to the clinical deployment of CAR-Tregs may limit the feasibility of comprehensive phenotypic and functional validation within the available timeframe.

### 3.5. Strength and Compatibility of Co-Stimulatory Domain

In the overall design of a CAR (chimeric antigen receptor), setting aside the selection of the binding domain and the common intracellular signalling domain CD3ζ, there are well-documented differences between the two main co-stimulatory domains used: CD28 and 4-1BB. To further explain, the 4-1BB exerts a weaker but persistent activation signal compared to CD28, which is rapid and short-lived [46]. This has not been conveyed as the consequence of CD28, which is also complicit in the early exhaustion of Tregs by the accelerated activation through hypersensitized receptors or lowered activation threshold and increased cytokine production [47]; albeit, the moderately weaker 4-1BB could retard or hinder the timely activation of the signal transduction pathway, incapacitating Treg’s suppressive role. Boroughs et al., 2019 verified this comparison in a controlled study, determining CD28-CD3ζ was preferred over 4-1BB-CD3ζ CAR Tregs for acute graft tolerance in a mice xenograft model [24]. Likewise, Imura et al., 2020 found that CD19-CAR carrying CD28 signalling domains were superior to 4-1BB for CAR-T expansion in vitro [19].

Despite known instability under inflammatory conditions, T cell exhaustion or phenotypic drift, CD28 co-stimulation has particularly useful qualities in the context of CAR signalling. Under controlled conditions, CD28 could promote stronger and sustained activation of canonical pathways, including IL-2R-STAT5 or CD28/CTLA-4 and FOXP3 maintenance [48,49,50]. Compared to CD28, 4-1BB co-stimulation favors generalized metabolic programs and effector functions that are less directed towards Treg phenotypes [21,51]. Moreover, previous evidence suggests that the Treg-specific chromatin landscape and transcriptional circuitry could buffer or resist the pro-effector signals generated by CD28, maintaining the stability of the Treg lineage [52]. While concerns around CD28-induced Treg fragility remain valid, the benefits of enhanced potency and persistence in transplantation models certainly outweigh the risk of low immunosuppressive properties, which ultimately lead to graft rejection. Moreover, CAR-Tregs are stringently characterized prior to infusion, ensuring manufacturing consistency (uniform subpopulation) and safety via preventing the risk of effector function conversion from the presence of specific biomarkers.

### 3.6. Biodistribution of CAR Tregs Key to Early Immune Desensitization

Antigen-specific CAR Tregs homing to the graft early is vital to prevent the influence of polyclonal Tregs from generating an immune response. However, it has become accepted that the CAR Tregs’ circulation also plays a critical role in the success of their immunosuppressive function. The CAR Tregs’ active draining into the immunological centres is a key factor in preventing GvHD and early desensitization of immunity. For CAR Tregs to initiate immune tolerance, especially in the context of transplantation or autoimmune disorders, they must gain access to these immune hubs to modulate the activity of other immune cells. These include APCs, DC, macrophages, and Teffs that could “make or break” successful transplants. Although both expressing and non-CD19-expressing Memory B cells (Bmem) and long-lived plasma cells (LLPC) are the other major culprits of transplant failures, their complete elimination poses the risk of compromised immunity long term [53]. In the Zhang et al., 2023 study, CD19-CAR T cells targeted CD19^+^ Bmem and LLPC, while the APRIL-CAR T cells targeted both CD19^+^ and CD19^−^ LLPC. The murine model was able to facilitate greater immune tolerance through combined CAR T (Combo-CART) vs. monotherapy, simultaneously reshaping the humoral immunity that remains functional but also tolerant to the allograft [32].

So far, none of the studies reported any challenges of CAR Tregs homing to the site of graft. Histopathology has revealed pulmonary congestion and hypercellularity of the spleen, consistent with IV administration [33]. From these studies reviewed, population or fragments of CAR Tregs were also found in the lymph nodes and bone marrow, indicative of their immune surveillance properties. Despite only a single administration of the FITC-H-2Dd-mABCAR Tregs, the accumulation of cells in lymphoid organs sustained for 3 weeks, proving long-term immunosuppressive function in the murine transplant model [23].

### 3.7. Complementary Action with Concomitant Therapy

Unlike CAR T cell therapy, CAR Tregs may not be making it as a first-line therapy in the foreseeable future. This is because first-line treatments must overcome numerous ethical, clinical, logistical, and risk-related hurdles. For example, long-term safety and efficacy data are still being gathered, where reports of adverse side effects are not well documented. On the other hand, first-line treatments should have proven efficacy in a significant number of patients and treatment settings. Tentatively, new therapeutic models such as CAR must be an adjunct therapy to prescribed immunosuppressants. These models will also require heavy scrutiny for underlying damage and/or dampening effects from first-line therapy.

In the latest findings, two studies showed promising results of CAR Tregs applied in tandem with immunosuppressants. Wagner et al., 2022 discovered that the haplo-identical heart transplant survived despite initial <14 days survival without treatment, the following treatment extended to >100 days for A2-CAR Tregs regardless of rapamycin treatment [30]. Supposedly, rapamycin spares Tregs but inhibits Tconv, which exceptionally contributes to its role as an immunosuppressant driver [54]. Rapamycin binds to FKBP12, which blocks the mammalian target of rapamycin complex 1 (mTORC1), leading to cell cycle arrest in T cells. On the other hand, low mTOR activity does not impede Treg activity, as it promotes their stability and enhances their suppressive function. Proics et al., 2023 also describe a CAR Treg model tolerable to tacrolimus, another common immunosuppressant for solid organ transplants [33]. Tacrolimus-FKBP complex specifically inhibits calcineurin, preventing transcription of the nuclear factor of activated T cells (NFAT) necessary for initiating T cell activation [55]. Through distinct pathways, both molecules actively suppress IL-2 production and sensitivity, indirectly retarding Treg persistence. Taken together, the success of these models suggests that patients on concomitant treatment do not need to discontinue their current treatment and may benefit from CAR Tregs simultaneously.

Beyond the scope of this review, a team recently validated a humanized and mutant IL-2 that selectively activates and expands endogenous Tregs in a murine model and cynomolgus monkey successfully [56]. Although both polyclonal and antigen-specific Tregs were expanded, the overall objective of an immunosuppressive microenvironment was achieved nonetheless. Thus, justifying further exploration of the mutant hIL-2. Perhaps, its effect could function synergistically with CAR Tregs to overcome viability issues and selective expansion of Tregs and not T cells in vivo. What is more, this novel cytokine could also be used as first-line therapy to modulate endogenous Tregs and simultaneously adjusted for the subsequent action of CAR Tregs if additional treatment is needed.

Owing to the fact that immune responses are primarily HLA-dependent in organ transplantation, mismatch of donor and recipient HLA alleles results in immediate graft rejection. Interestingly, the discovery of HLA-independent TCR (HIT) receptors is emerging as a potential way to bypass this issue and possibly shine light on mismatched donor transplants [57]. Since HIT receptors recognize a wider range of antigens, beyond the HLA complex, they could be designed to recognize donor-specific antigens universally expressed across different organ types. A recent study demonstrated that HIT receptors are consistently able to recognize target cells with high antigen sensitivity, despite the low abundance of target antigens [58]. Hence, combining HIT receptors with CAR-Tregs could answer how future organ transplants could be performed with less dependence on HLA compatibility.

Although CAR Treg therapy holds great promise, several obstacles continue to impede its clinical application. A key issue lies in preserving the stability and suppressive identity of Tregs following genetic modification, large-scale expansion, and in vivo administration. Under certain conditions, cells may lose FOXP3 expression and transform into a pro-inflammatory phenotype, potentially causing harmful immune responses rather than suppression [34]. The production process is also demanding, requiring compliance with stringent GMP standards to ensure cell purity, safety, and functional consistency—factors that could restrict scalability and raise manufacturing costs [59]. Additional concerns include the risk of off-target activity or unintended immune effects, particularly when donor antigens are expressed in non-target tissues. Furthermore, questions remain regarding the long-term persistence, survival, and regulatory function of CAR Tregs once infused into patients. Nevertheless, progress in areas such as synthetic biology, vector optimization, and strategies to stabilize the Treg phenotype is steadily advancing the field, bringing CAR Treg therapy closer to clinical reality as a targeted solution for preventing allograft rejection.

## 4. Conclusions

In summary, there is adequate evidence to support the proof-of-concept for CAR Tregs as an immunomodulatory therapy. Since it is in the relatively early stages of a working prototype, more research and experiments are needed. The design parameters of CAR Tregs are abundant but lack validation, making the interpretation subjective. The shortcomings of this review include the inability to determine the large bioprocessing and clinical translation capacity of CAR Tregs. This will be a critical factor in ensuring access to life-saving alternatives when the likelihood of success with organ transplants is low due to a high risk of allograft rejection.

## Figures and Tables

**Figure 1 biomedicines-13-01757-f001:**
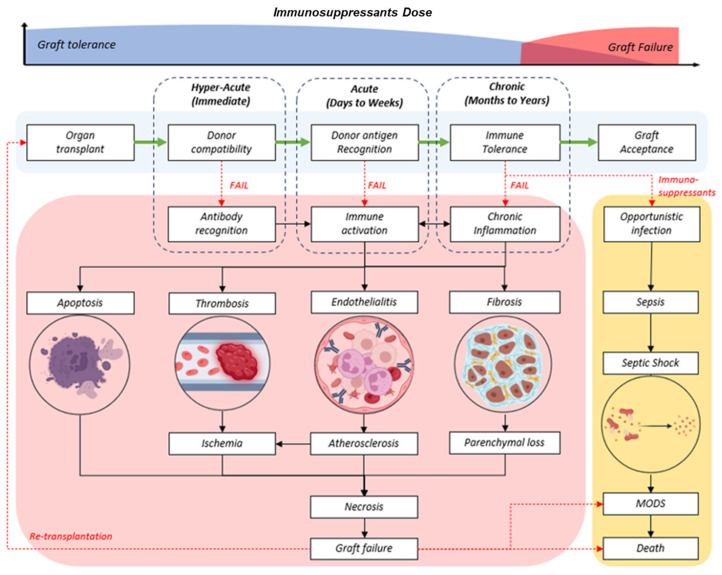
(**Top**) Trend of life-long immunosuppressants in graft tolerant (blue) or increased dose during chronic, delayed, or failed transplants (red). (**Bottom**) Generalized flow chart of successful graft progress (light blue) and the phases of rejection (red) at different phases (dotted box) in order of duration post-transplant: hyperacute, acute, and chronic.

**Figure 2 biomedicines-13-01757-f002:**
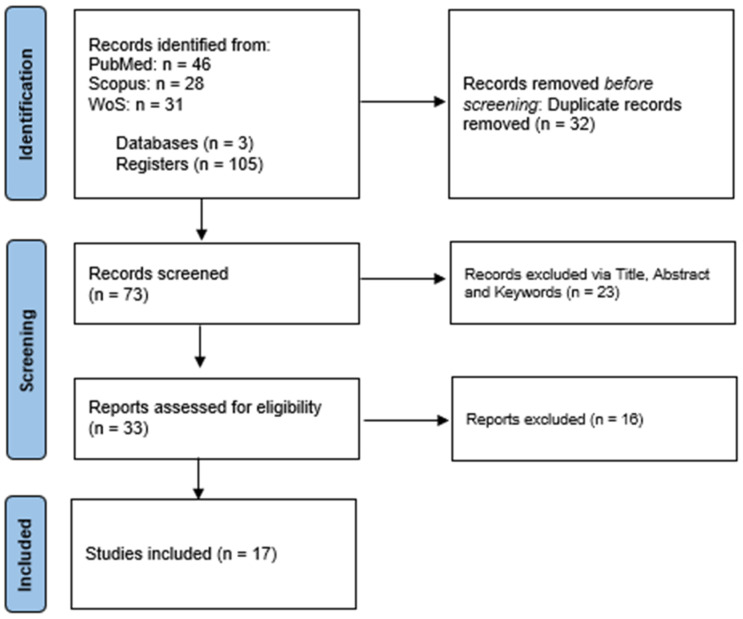
PRISMA flow diagram.

**Figure 3 biomedicines-13-01757-f003:**
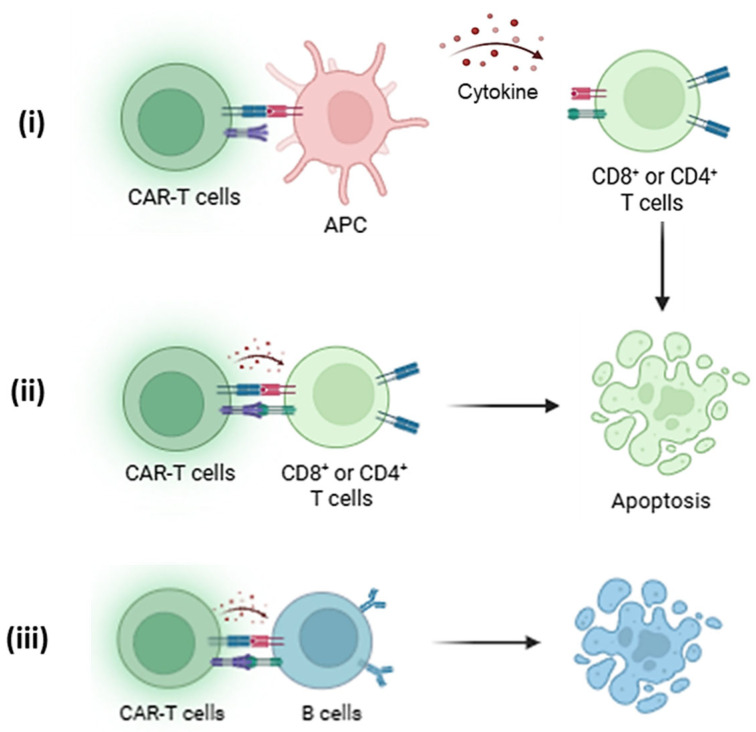
(**i**) HLA-A2-specific CAR-Tregs interact with antigen-presenting cells (APCs) by recognizing HLA-A2 on their surface, leading to the suppression of co-stimulatory molecule expression and promoting a tolerogenic phenotype through the release of IL-10 and CTLA-4-mediated signalling [14]. This dampens APC-mediated activation of effector T cells (Teffs), which are further suppressed by (**ii**) CAR-Tregs through cytokine deprivation, inhibitory cytokines, and contact-dependent mechanisms. Thus, leading to deactivation or apoptosis of Teffs [17]. (**iii**) Additionally, CAR-Tregs can modulate B cell activity by reducing alloantibody production within the HLA-A2^+^ immune environment or trigger apoptosis, contributing to broader immune tolerance [37].

**Table 1 biomedicines-13-01757-t001:** Preclinical model of induced GvHD or tissue transplant, dose of CAR T model, and in vivo analysis (see end of document).

First Author & Year	Animal Model	Mode of Induced GvHD	CAR T Cell Model and Administered Dose	Results or Outcome
Macdonald et al., 2016 [22]	8 to 12-week-old NSG mice (*n* = 4/group).	hPBMC (1 × 10^7^ cells/mice) injection from 3 to 4 healthy donors.	HLA-A2-CAR CD4^+^ Tregs at 0.5 or 1 × 10^7^ cells, once per study.	Survival of mice treated with A2-CAR Tregs was significantly higher compared to HER2-CAR Tregs. Mice treated with A2-CAR Tregs had a delayed onset of GVHD and lower weight loss. In blood samples of treated mice, A2-CAR Tregs showed higher FOXP3 expression and persisted longer (2 weeks) than HER2-CAR Tregs.
Noyan et al., 2017 [17]	8 to 10-week-old NRG mice (*n =* 3 to 7/group).	Human skin transplant followed by hPBMC (7.5 × 10^6^ cells/mice) injection from healthy donors.	HLA-A2-CAR CD4^+^ Tregs at 1 × 10^6^ cells, once per study.	A2-CAR Tregs significantly reduced delayed-type hypersensitivity (DTH) response compared to non-transduced Tregs. The ear-swelling response was significantly lower by 70% (*p* < 0.01) than polyclonal Tregs. In immune-reconstituted humanized NRG mice, A2-CAR Tregs completely prevented the rejection of HLA-A*02-positive target cells, unlike control CAR Tregs or polyclonal Tregs.
Boardman et al., 2017 [18]	10 to 11-week-old BRG mice (*n =* 2 to 3/group).	Human skin transplant followed by hPBMC (5 × 10^6^ cells/mice) injection from 16 healthy donors.	HLA-A2-CAR CD4^+^ Tregs with or without CD28-CD3ζ signalling domain at 1 × 10^6^ cells, once per study.	Unlike CAR-modified Teffs, CAR Tregs did not exhibit cytotoxicity against HLA-A2+ cells. They produced low levels of IFN-γ and high levels of IL-10 (*p* < 0.05), contributing to an immunosuppressive environment. In a human skin xenograft model, CAR Tregs provided superior protection against alloimmune-mediated skin graft rejection compared to polyclonal Tregs.
Pierini et al., 2017 [23]	2 to 4-month-old BALB/c mice (*n =* 5/group).	GvHD induced by Tcons (1 × 10^6^ cells/mice) injection from allogeneic donor mice. Allogeneic islet transplant from FVB/N mice.	FITC-H-2D^d^-mABCAR CD4^+^ Tregs or T cells at 0.5 or 1 × 10^6^ cells, once per study.	Survival probability and GvHD score of mice treated with mAbCAR Tregs were significantly higher compared to untreated mice (*p* < 0.001). The mAbCAR Tregs increased islet allograft survival compared to control Tregs, also indicated by the significant reduction in CD8+ T cell infiltration (*p* < 0.05).
Boroughs et al., 2019 [24]	8 to 10-week-old NSG mice (*n =* 5/group).	Human skin transplant from 4 healthy donors.	CD19-28ζ or EGFR-28ζ CAR CD4^+^ Tregs or CD8^+^ Teff cells or both at 2 × 10^6^ cells each, once per study.	Although both models were superior to Tconv, the CD28 CAR-Tregs expressed significantly less tissue destruction (*p* < 0.0001) and CD8+ T cell infiltration of graft (*p* < 0.0001) compared to 4-1BB CAR-Tregs. Thus, it suppressed Teff-mediated tissue damage, leading to prolonged graft survival in a skin xenograft model.
Dawson et al., 2019 [25]	8 to 12-week-old NSG mice (*n =* 4/group).	hPBMC (8 × 10^6^ cells/mice) injection from 4 healthy donors. Allogeneic skin graft (NSG-A2 mice) and xenogeneic (HLA-A2^+^) from healthy donors.	HLA-A2 CAR Tregs (4 × 10^6^ cells/mice), once per study.	In the xenogeneic GvHD model, the A2-CAR Tregs improved survival (*p* < 0.001) and reduced human CD45+ cell engraftment by ~60% (*p* < 0.01) compared to control groups. Additionally, the bioluminescence imaging revealed lesser formation of keratinocytes and involucrin destruction, prompting better survival of the graft in the A2-CAR Treg-treated mice.
Bézie et al., 2019 [26]	8 to 12-week-old NSG mice (*n =* 3 to 9/group).	hPBMC (1.5 × 10^7^ cells/mice) injection from 4 healthy donors. Human skin transplant from healthy donors.	HER2- or HLA-A2 CAR CD8^+^ Tregs (Ratio of 1:1 or 3:1 of PBMC:Tregs), once per study.	The A2-CAR CD8+ Tregs significantly inhibited GvHD compared to HER2 CAR Tregs at 100% and 25% survival rate (*p* < 0.01), respectively. The A2-CAR CD8+ Tregs prolonged skin graft survival for over 100 days (*p* < 0.001). The effects of the immunotherapy also behaved in a dose-dependent manner.
Dawson et al., 2020 [27]	8 to 12-week-old NSG mice (*n =* 3 to 9/group).	HLA-A2^+^ PBMC (10 × 10^7^ cells/mice)	CD28wt-, CD28mut, ICOS, CTLA-4wt, CTLA-4mut, PD-1, OX40, GITR, 4-1BB, or TNFR2 CAR CD4^+^ Tregs at 2.5 or 5.0 × 10^6^ Tregs, once per study.	Among the 10 CAR models, exploring different co-stimulatory domains, the CD28wt produced the best survival outcome for the GvHD animal model. Furthermore, the results show significantly higher FoxP3^+^ and Helios expression of cells in circulation—indicating CAR Treg stability and persistence in vivo.
Sicard et al., 2020 [14]	8 to 16-week-old C57BL/6 mice (*n =* 5 to 11/group).	Skin grafts from syngeneic and/or HLA-A2^+^ mice.	HER2- or HLA-A2 CAR CD4^+^ Tregs at 1 × 10^6^ (equivalent to 30–50 × 10^6^/kg) Tregs, once per study.	The dsCAR-Tregs (14 days) significantly median survival times (*p* < 0.0001) for the grafts when compared to irrCAR-Tregs (8.5 days) and PBS-treated control (8.0 days). Additionally, dsCAR-Tregs significantly (*p* < 0.05) reduced the generation of DSAs and the number of DSA-secreting cells in naive recipients, compared to irrCAR-Tregs.
Imura et al., 2020 [19]	8 to 11-week-old NSG mice (*n =* 4 to 7/group).	hPBMC (5 × 10^6^ cells/mice) injection from 5 healthy donors.	HER2- or CD19- CAR CD4^+^ Tregs at 2 × 10^6^ cells at 4 to 6 hrs or 7 days after.	In vivo, the CD19-CAR Tregs significantly reduced serum levels of IgG and IgM levels (*p* < 0.05), suppressed CD20+ B cell expansion, and reduced GvHD scores (*p* < 0.01) more effectively than polyclonal Tregs. Compared to CAR CD8^+^ T cells, the CAR Tregs had significantly lower (*p* < 0.05) IL-6 levels and greater Tregs, preventing further weight loss and exacerbating GvHD.
Lee et al., 2022 [28]	Unspecified C57BL/6J mice (*n =* 3/group).	Heart transplant from healthy BALB/c mice.	C4d-CAR CD4^+^ Tregs at 1 × 10^6^ cells, once per study.	In a mouse model of ABO-incompatible heart transplantation, recipients treated with anti-C4d CAR Tregs had significantly prolonged (*p* < 0.05) allograft survival compared to those treated with control CAR Tregs. Histological analysis showed that anti-C4d CAR Tregs reduced inflammation in the heart graft, as evidenced by lower expression of pro-inflammatory cytokines IFN-γ and TNF-α.
Muller et al., 2021 [29]	7 to 8-week-old NSG mice (*n =* 2 to 6/group).	hPBMC (5 × 10^6^ cells/mice) injection from deceased, de-identified donors. 500 mouse islets or 3000 human islet equivalents (IEQs) transplanted.	HLA-A2-TCR^deficient^ CD4^+^ Tconv or Tregs at 2.5 × 10^6^ cells/mice, once per study.	The A2-CAR Tregs demonstrated the antigen specificity of the CAR by significantly improving (*p* < 0.01) the survival of HLA-A2+ mice compared to control mice, which expired approximately 13 days post-transplant. Also, the A2-CAR Tregs were trafficked specifically to the graft site and prevented the onset of rejection for up to 11 days compared to Tconv after 6 days (*p* < 0.001).
Wagner et al., 2022 [30]	Unspecified C57BL/6J mice (*n =* 3 to 7/group).	Heart transplants from single A2-mismatch, haplo-mismatch, and syngeneic donor mice.	HLA-A2 CAR CD4^+^ Tregs at 1 × 10^6^ cells/mice at 1, 2, or 4 × 10^6^ cells/mice, once per study at Day 1 or 2 post-transplant.	The administration of A2-CAR Tregs increased graft survival (*p* < 0.05) 99 and 35 days, respectively, compared to without treatment at 23 days. Despite the incremental doses, there were no significant differences reported. With or without rapamycin, the strongly immunogenic, haplo-mismatched heart transplantation model survival was extended from 14 days to 100 days (*p* < 0.05).
Rosado-Sánchez et al., 2023 [31]	10 to 14-week-old C57BL/6 mice (*n =* 3 to 13/group).	Skin grafts from syngeneic and/or HLA-A2+ mice.	HLA-A2 with or without CD28 co-stimulatory domain of CAR CD4^+^ Tregs at 1 × 10^6^ cells/mice, once per study.	CAR Tregs with TNFR family domains (OX40, 4-1BB) did not affect DSA levels, causing failed graft engraftment. Conversely, CD28 and PD-1 CAR Tregs reduced DSA levels, which improved median graft survival (*p* < 0.01) to 20 days and 19.5 days, respectively, from 14 days untreated. Interestingly, the absence of the CD28 co-stimulatory domain on the A2-CAR Treg still enabled allograft tolerance, compensated by endogenous APCs or dendritic cells.
Zhang et al., 2023 [32]	8 to 12-week-old C57BL/6J mice (*n =* 3/group).	Skin graft from BALB/cJ mice. Islet transplant from BALB/cJ mice.	CD19 and/or APRIL CAR CD8^+^ T cells at no specified dose.	The combination of CART-19 and APRIL-CAR (Combo-CART) was highly effective at depleting B cells and PCs in the bone marrow and spleens by approximately 70% (*p* < 0.05) compared to CART-19 monotherapy. Similarly, the Combo-CART-protected islet allograft up to 30 days (*p* < 0.01), comparable to the non-sensitized group receiving standard treatment. Interestingly, the eventual depletion of CAR T cells’ activity and resurgence of B cells that returned levels of IgGs to normal but without DSAs had likely displaced endogenous B cells and PCs with graft-tolerant versions.
Proics et al., 2023 [33]	7 to 8-week-old NSG mice (*n =* 3 to 10/group).	hPBMC (5 × 10^6^ cells/mice) from 2 healthy donors. Human skin transplant from a healthy donor.	HLA-A2 or TX200-TR101 CAR CD4^+^ Tregs at 1 × 10^6^ (equivalent to 30–50 × 10^6^/kg) Tregs, once per study.	The group treated with TX200-TR101 Tregs exhibited significantly lower (*p* < 0.05) GvHD scores (0.6) and improved survival (*p* < 0.05) compared to control groups. TX200-TR101 Tregs localized at human skin grafts and significantly improved delayed rejection (*p* < 0.001), resulting in a median survival rate of 35 days compared to 16 days in the control group.
Lamarche et al., 2023 [21]	8 to 12-week-old NSG mice (*n =* 5 to 7/group).	hPBMC (6 × 10^6^ cells/mice) from 3–5 healthy donors.	CD19 or GD2 CAR CD4^+^ Tregs at 3 × 10^6^ Tregs, once per study.	Despite demonstrating immunosuppressive capacity in vitro, TS-CAR Tregs failed to confer protection against GvHD in vivo. This loss of function was associated with Treg exhaustion, as evidenced by the increased graft immune infiltration and tissue loss. The survival outcomes were comparable to the untreated group and significantly worse than those of animals treated with untransduced Tregs (*p* < 0.01), indicating that Treg exhaustion critically impairs their regulatory function in the allogeneic setting.
Henschel et al., 2023 [34]	Unspecified NRG mice (*n =* 3 to 6/group).	hPBMC (5 × 10^5^ cells/mice) from 2–4 healthy donors, three times per week.	HLA-A2 or HLA-A2-FOXP3 CAR CD4^+^ CD45RA^+^ Tregs at 5 × 10^5^ Tregs, once per study.	The FOXP3-CAR Tregs effectively prevented (*p* < 0.05) immune-mediated destruction of allogeneic target cells compared to the control group. Even in severely high inflammatory and IL-2-deprived conditions, the FOXP3-CAR Tregs retained ~75% (*p* < 0.001) of their FOXP3 expression and nearly two-fold more compared to the control group.

**Table 2 biomedicines-13-01757-t002:** CAR design, transduction efficiency, and in vitro phenotypic analysis (see end of document).

First Author & Year	CAR Structure					CAR Generation Efficiency or Viability (%)	Implication of CAR Design
	Binding	Hinge	Transmembrane	Co-Stimulatory	Signal		
Macdonald et al., 2016 [22]	αHLA-A2 or αHER2	CD8α-CD28	CD28	-	CD3ζ	86–96%	A2-CAR Tregs was successfully generated with a consistent phenotype of high expression of FOXP3, CD25, and CTLA-4 (*p* < 0.05). Through mixed lymphocyte reactions, A2-CAR Tregs greatly suppressed the proliferation of HLA-A2+ T cells compared to HER2-CAR Tregs (*p* < 0.001). Other regulatory markers like suppressive markers, CTLA-4 and GARP, were greatly expressed (*p* < 0.05) in the CAR Tregs than the TCR-stimulated Tregs (*p* < 0.05).
Noyan et al., 2017 [17]	αHLA-A2	CD8α	CD28	-	CD3ζ	55–95%	No significant alterations to the regulatory phenotype of A2-CAR Tregs as FOXP3, CCR7, CD45RO, CD45RA, and CD39 expression remained stable. A2-CAR Tregs produced higher suppression of Teff proliferation compared to control CAR Tregs. Even at a low (1:64) ratio, the CAR Tregs continued to suppress >60% of Teff proliferation (*p* < 0.01).
Boardman et al., 2017 [18]	αHLa-A2	CD28	CD28	-	CD3ζ	40–80%	After transduction, eGFP, FOXP3, and CTLA-4 remained highly expressed by up to >90% in CAR Tregs. CAR-mediated antigen-specific suppression was confirmed through lesser suppression of effector T cells (*p* < 0.01) when further cocultured with HLA-A2^+^ APCs. In spite of that, the CAR Treg did not elicit cytotoxic effects against non-target HLA-A2^+^ cells. Modified secretion of lower IFN-γ and higher IL-10 was also detected from the CAR Tregs (*p* < 0.05).
Pierini et al., 2017 [23]	αFITC-H-2D	CD28	CD28	-	CD3ζ	30%	A modular CAR system (mAbCAR) that specifically binds FITC and was successfully expressed in both T cells and Tregs. Activation leads to T cell activation (CD69 and CD25 expression) and the generation of an effector memory phenotype (CD44+CD62L-). On the other hand, FITC-stimulated mAbCAR Tregs retained their suppressive function through high FOXP3 expression and increased CD69 and PD-1 expression (*p* < 0.05), preventing GvHD while maintaining immune tolerance.
Boroughs et al., 2019 [24]	αCD19 or αEGFR	CD8	CD3ζ	CD28 or 4-1BB	CD3ζ;	>50%	Although no significant changes occurred for FOXP3 expression, CD69 expression and IL-10 secretion were significantly higher (*p* < 0.05 and *p* < 0.01, respectively) in the CD28 CAR-Tregs vs. 4-1BB CAR-Tregs. The pro-inflammatory cytokines TNF-α, GM-CSF, IL-2, and IFN-γ) were also significantly reduced (*p* < 0.05) in the CD28 CAR-Tregs groups. As a result, CD28 CAR-Tregs suppressed Teff proliferation by 75%, while 4-1BB CAR-Tregs showed only 20% suppression (*p* < 0.01).
Dawson et al., 2019 [25]	αHLA-A2 or αHER2	CD8α	CD28	-	CD3ζ	Not specified	From the twenty variants of hA2-CARs generated, the ten selected had significantly higher (*p* < 0.01) CD69 and CD71 expression than non-transduced controls. Additionally, only seven hA2-CARs showed robust activation of Tregs upon stimulation with HLA-A2–positive cells. Thus, immunosuppression of effector T cells was 40% more than control Tregs (*p* < 0.01). Also, the cross-reactive HLA-A alleles were reduced up to 90% (*p* < 0.05) in hA2-CAR Tregs compared to mA2-CAR, highlighting the potential significance of xenogeneic factors influencing immunity.
Bézie et al., 2019 [26]	αHLA-A2 or αHER2	-	CD28	-	CD3ζ	20%	A2-CAR expression was stable after 14 days of culture, retaining its regulatory phenotype after CAR expression. The transduced cells expressed FOXP3, IL-10, and IFN-γ, without gaining cytotoxic markers such as perforin and FASL. Unlike A2-CAR CD8+ Teffs, A2-CAR Tregs did not cause cytotoxicity (*p* < 0.01) in HLA-A*02+ endothelial cells (ECs). The A2-CAR also suppressed (*p* < 0.01) effector T cell proliferation by up to 80%, succeeding HER2-CAR Tregs and non-transduced Tregs.
Dawson et al., 2020 [27]	αHLA-A2 or αHER2	-	CD28, ICOS, OX40, GITR, 4-1BB, or TNFR2	CD28, CD28 (Y173F), ICOS, CTLA-4, CTLA-4 (Y165G), PD-1, OX40, GITR, 4-1BB, or TNFR2.	CD3ζ	>50%	CD28wt and CD28mut stimulated the highest CD69, CD71, LAP, CTLA-4, and GARP expression following antigen-specific stimulation (*p* < 0.01). CD28wt, 4-1BB, and TNFR2 CARs induced proliferation, but only CD28wt preserved FOXP3 and Helios expression after 12 days of CAR stimulation. Helios loss correlated with instability, confirmed by increased FOXP3 TSDR methylation in TNFR2-CAR Tregs (*p* < 0.01).
Sicard et al., 2020 [14]	αHLA-A2 or αHER2	CD8a	CD28	-	CD3ζ	75%	Transduced donor-specific (ds) Tregs of anti-HLA-A2-specific CAR were successful, maintaining expression of FoxP3, CD4, and regulatory markers. The dsCAR-Tregs suppressed (*p* < 0.05) the proliferation of HLA-A2+ antigen-presenting cells (APCs) and CD4+ effector T cell proliferation in vitro compared to irrelevant CAR-Tregs (irrCAR-Tregs).
Imura et al., 2020 [19]	αCD19 or αHER2	CD28	CD28	-	CD3ζ	50%	The generated CD19-CAR Tregs maintained proliferation capacity and expression of regulatory markers, including FoxP3, Helios, and CTLA-4, similar to polyclonal Tregs. However, the former showed higher levels (*p* < 0.01) of anti-inflammatory cytokines like IL-10 and lower levels of IFN-γ and IL-2. The CD19-CAR Tregs also reduced (*p* < 0.01) B cell proliferation by up to 70% and IgG production (*p* < 0.05).
Lee et al., 2022 [28]	αC4d	CD8	CD28	-	CD3ζ	Not specified	There was no significant difference in the expression of regulatory markers (FoxP3, CD25, CTLA-4, and GITR) between anti-C4d CAR Tregs and NT Tregs. However, C4d-CAR Tregs showed enhanced activation by increased CD69 expression and IL-10 secretion compared to both control CAR and NT Tregs (*p* < 0.01). When cocultured with effector T cells, the anti-C4d CAR Tregs suppressed T cell proliferation significantly more (*p* < 0.05) than NT Tregs.
Muller et al., 2021 [29]	αHLA-A2	IgG4	CD28	-	CD3ζ	50%	Using CRISPR/Cas9, the endogenous TCR was successfully knocked out, and the A2-CAR construct was integrated into the TRAC locus at 85% and 91% efficiency (*p* < 0.01), respectively. The TCR-deficient A2-CAR Tregs maintained high levels of regulatory markers such as FoxP3, Helios, and CD25 (*p* < 0.05). A2-CAR Tregs specifically suppressed the proliferation of HLA-A2+ effector T cells by 80% (*p* < 0.01) compared to polyclonal Tregs.
Wagner et al., 2022 [30]	αHLA-A2	CD8	CD28	-	CD3ζ	60–80%	The A2-CAR Tregs exhibited enhanced proliferation by 8- to 16-fold higher (*p* < 0.0001) compared to non-transduced Tregs. Furthermore, their immunosuppressive function in the presence of HLA-A2-expressing cells, marked by CD69 and CD25 activation, was upregulated (*p* < 0.01).
Rosado-Sánchez et al., 2023 [31]	αHLA-A2	CD28	CD28	ICOS; -PD-1; -GITR, and -TNFR family proteins	CD3ζ	70%	Eight different HLA-A2–specific CAR variants with distinct co-stimulatory domains, Tregs expressing the CD28-based CAR demonstrated the strongest proliferative capacity (*p* < 0.001) in vitro when cocultured with HLA-A2–expressing cells. As a result, immunoregulatory response, including cytokine secretion like IL-10, was highest in CD28-CAR Tregs (*p* < 0.01) vs. other variants.
Zhang et al., 2023 [32]	αCD19 or αAPRIL	-	CD28	-	CD3ζ	Not specified	Among the cohort of five multiple myeloma patients with pre-existing DSAs, Combo-CART therapy (CART-BCMA and CART-19) significantly reduced allo-antibodies in three subjects by 47% to 97%. The decline in allo-antibodies was statistically significant (*p* < 0.05) and was accompanied by total reduction of IgM, IgA, and IgG levels (*p* < 0.05).
Proics et al., 2023 [33]	αHLA-A2 or TX200-TR101	-	CD28	-	CD3ζ or without	90%	TX200-TR101 Tregs retained a high expression (>89%; *p* < 0.05) of Treg markers, such as FOXP3 and CD25, and were stable after expansion. TX200-TR101 Tregs achieved 52.7% suppression (*p* < 0.05) compared to the 19.2% suppression of Tconv by control Tregs.
Lamarche et al., 2023 [21]	αCD19 or αGD2	-	CD28	-	CD3ζ	75%	Compared to untransduced or non-tonic-signalling (non-TS) groups, the proportion of TS-CAR Tregs expressing inhibitory receptors—the likes of PD-1, TIM-3, LAG-3, GITR, and 4-1BB—was significantly higher (*p* < 0.05). Higher glycolysis and acidification (*p* < 0.05) were also observed, relevant to the exhaustive state of TS-CAR Tregs. Despite that, it remained highly suppressive (>70%) in vitro against CD4+ and CD8+ effector T cells (*p* < 0.01) vs. UT and non-TS-CAR Tregs.
Henschel et al., 2023 [34]	αHLA-A2 with or without -FOXP3	-	CD28	-	CD3ζ	75%	CAR modification confirmed a 40% higher level of FOXP3 expression vs. control CAR-Tregs (*p* < 0.001). Supra-expression of FOXP3 resulted in a low-level cytokine secretion profile: TNF-α, IFN-γ, IL-2, and IL-17 (*p* < 0.0001); suppression of T-effector cells by 70% (*p* < 0.01); retention of FOXP3^+^ subpopulation under IL-2 deprived conditions (*p* < 0.001).

## Data Availability

Not applicable.

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
