# Peer review of "Revolutionizing Allogeneic Graft Tolerance Through Chimeric Antigen Receptor-T Regulatory Cells"

_biomedicines, 2025, doi:10.3390/biomedicines13071757_

Round 1

Reviewer 1 Report

Comments and Suggestions for Authors

Title: Revolutionizing Allogeneic Graft Tolerance through Chimeric 2 Antigen Receptor-T regulatory Cells

Organ transplantation, often leads to graft rejection due to immune responses.  CAR-Tregs are engineered to suppress immune responses more specifically, offering an alternative to lifelong immunosuppressants. This review focuses on the preclinical progress of CAR-Treg therapy in the context of organ and allogeneic transplantation.

Comments:

HLA-independent T-cell receptors (HIT receptors) have also shown promising new applications. Including a brief discussion of these advances would strengthen the article and provide a more comprehensive overview of emerging T cell therapies.

Since that ex vivo expansion protocols may contribute to tonic signaling and subsequent exhaustion in CAR-Tregs. It would strengthen the review to include studies that compare different expansion strategies, also today some people used G-REX plate to expand the T-reg cell instead of regular ones.

The manuscript suggests that FOXP3 expressions should be maintained above 90% as a benchmark for functional CAR-Tregs. However, it is well-documented that prolonged in vitro culture and expansion can lead to the loss of FOXP3 expression in a significant proportion of Tregs, potentially compromising their stability and suppressive function. It would be valuable for the authors to clarify: At what time point(s) was FOXP3 expression assessed in the studies reviewed - immediately after transduction, after expansion, or at the time of functional testing? Also, transducing T-reg cells is not an easy task because they lose their function quickly due to their plasticity, making it difficult to differentiate between Trge and CD4 cells.

Given this loss over time, what would be the optimal window for phenotypic and functional validation of CAR-Tregs to ensure accurate evaluation?

Addressing these points would improve the manuscript’s utility for researchers developing clinically relevant CAR-Treg protocols.

Reviewer 2 Report

Comments and Suggestions for Authors

This study provides a systematic literature review of CAR-Treg, an innovative method to prevent immunological diseases that may occur after long-term transplantation. Here are some suggestions for revision. We hope these suggestions will help you improve the quality of your paper.

  1. This manuscript has minor but noticeable editorial issues. There are two figures, shown as "Figure 2". This duplication should be corrected to prevent confusion of references and citations in the text.

  1. Section 3.5 states that CD28-based CAR-Tregs are superior in terms of transplantation habits to 4-1BB-based CAR-Tregs, but there is little explanation of the mechanism. Given the early concerns of CD28-related shortages and FOXP3 instability, the authors must clearly explain why CD28 was preferred and why the benefits of CD28 in a Treg-specific situation may outweigh such risks.

  1. Throughout the paper, conventional T cells (T conv), . Conventional Tregs (Tconvs) are changed to one "s" to give confusion of meaning. This may cause a lot of confusion for readers.

  1. This paper is different from a general review in that it analyzes only a specific range of papers and therefore cannot include a wide variety of research results. This is understandable. However, if this is the case, it would be necessary to provide more detailed coverage of the mechanisms and basic concepts that are important to address in this review

Round 2

Reviewer 1 Report

Comments and Suggestions for Authors

Dear Editor, Thank you for the opportunity to review the revised version of the manuscript titled "Revolutionizing Allogeneic Graft Tolerance through Chimeric Antigen Receptor-T Regulatory Cells."
The authors have addressed the reviewers' comments appropriately, and the revisions have significantly improved the clarity and scientific rigor of the manuscript. I recommend acceptance in its current form.